# Niclosamide and Pyrvinium Are Both Potential Therapeutics for Osteosarcoma, Inhibiting Wnt–Axin2–Snail Cascade

**DOI:** 10.3390/cancers13184630

**Published:** 2021-09-15

**Authors:** Young Yi, Young Mi Woo, Kyu Ho Hwang, Hyun Sil Kim, Sang Hyeong Lee

**Affiliations:** 1Department of Orthopedics, Inje University Seoul Paik Hospital, Seoul 04551, Korea; J5329@paik.ac.kr; 2Department of Oral Pathology, Oral Cancer Research Institute, Yonsei University College of Dentistry, Seoul 03722, Korea; marzo15th@hanmail.net (Y.M.W.); tinogo@yuhs.ac (K.H.H.); 3Department of Orthopedics, Dongguk University Ilsan Hospital, Goyang-si 10326, Gyeonggi-do, Korea

**Keywords:** osteosarcoma, epithelial–mesenchymal transition, niclosamide, pyrvinium, Wnt, Snail, Axin2

## Abstract

**Simple Summary:**

Epithelial–mesenchymal transition (EMT) regulated by Wnt signaling is known as a key mechanism of cancer progression. Although evidence has suggested that the oncogenic Wnt signaling pathway and EMT program are important in the progression of osteosarcoma, there is no known therapeutic drug targeting EMT for osteosarcoma. We investigated whether Axin2, an important EMT target, could be a suitable molecular target and biomarker for osteosarcoma. Furthermore, we showed that both niclosamide and pyrvinium target Axin2, and effectively induce EMT reversion in osteosarcoma cell lines. Our findings suggest an effective biomarker and potential EMT therapeutics for osteosarcoma patients.

**Abstract:**

Osteosarcoma, the most common primary bone malignancy, is typically related to growth spurts during adolescence. Prognosis is very poor for patients with metastatic or recurrent osteosarcoma, with survival rates of only 20–30%. Epithelial–mesenchymal transition (EMT) is a cellular mechanism that contributes to the invasion and metastasis of cancer cells, and Wnt signaling activates the EMT program by stabilizing Snail and β-catenin in tandem. Although the Wnt/Snail axis is known to play significant roles in the progression of osteosarcoma, and the anthelmintic agents, niclosamide and pyrvinium, have been studied as inhibitors of the Wnt pathway, their therapeutic effects and regulatory mechanisms in osteosarcoma remain unidentified. In this study, we show that both niclosamide and pyrvinium target Axin2, resulting in the suppression of EMT by the inhibition of the Wnt/Snail axis in osteosarcoma cells. Axin2 and Snail are abundant in patient samples and cell lines of osteosarcoma. The treatment of niclosamide and pyrvinium inhibits the migration of osteosarcoma cells at nanomolar concentrations. These results suggest that Axin2 and Snail are candidate therapeutic targets in osteosarcoma, and that anthelminthic agents, niclosamide and pyrvinium, may be effective for osteosarcoma patients.

## 1. Introduction

Osteosarcoma is the most common malignant bone tumor in adolescents [1]. The relative five-year survival rate of adolescents is approximately 60%, but patients with metastatic lesions have a much poorer five-year survival rate [1]. The treatment of choice for osteosarcoma is en bloc resection of the cancer. Neoadjuvant chemotherapy for osteosarcoma with various chemotherapeutic agents has been attempted since the 1970s, but it has disadvantages of side-effects such as toxicity in various organs, including bone marrow suppression [2]. Moreover, it is often difficult to achieve complete remission for osteosarcoma with chemotherapy due to resistance to chemotherapeutic agents. Therefore, in vivo it is necessary to develop a new anticancer drug for osteosarcoma that can reduce the side-effects of current anticancer drugs and have excellent effects.

The invasive growth and metastatic process of cancer cells depends on the epithelial–mesenchymal transition (EMT), a complex biological mechanism. In addition to invasion, EMT in cancer is recently believed to be involved in resistance to cell death, resistance to chemotherapy, immunotherapy and radiation therapy, and stem cell properties [3,4,5,6,7]. It is known that EMT is induced by Snail, a transcriptional repressor of E-cadherin [8,9]. The findings that the Snail-dependent EMT program is regulated by the major oncogenic Wnt signaling pathway has provided a new molecular basis for understanding cancer progression [10,11]. Among the Wnt downstream target genes, only Axin2 appears to act as a regulator of Wnt signaling [12,13,14]. Axin2, a scaffolding protein of GSK3β, induces EMT by regulating the nuclear export of GSK3β and nuclear stability of Snail [11]. In various types of human cancer and precancer tissue, Axin and Snail protein are consistently co-localized, which supports that Axin2 serves as an oncogenic role by inducing Snail-mediated EMT [11,15].

Niclosamide and pyrvinium are FDA-approved anthelmintic drugs effective for tapeworm infections and pinworm infection, respectively [16,17]. Both niclosamide and pyrvinium have been reported to inhibit Wnt/β-catenin pathways, and elicit antitumor responses in various human cancer cells of ovarian cancer, prostate cancer, breast cancer, and colon cancer [18,19,20,21,22,23]. It has recently been identified that the direct molecular target for niclosamide is the Axin2–GSK3β interaction site, and niclosamide decreases β-catenin and Snail abundance and induces mesenchymal to epithelial reversion [24,25].

The components of the Wnt signaling pathway and EMT-related transcription factors, such as Snail, Zeb, and Twist, are consistently overexpressed in osteosarcoma [26,27]. Several reports have demonstrated a hyperactivated Wnt pathway in osteosarcoma patients related to distant metastases and poor survival rates [28,29,30]. Although evidence has suggested that the Wnt signaling pathway and EMT program are important in the progression of osteosarcoma, the suitable molecular target and effective therapeutic drug based on EMT for osteosarcoma remains unclear. In this study, we found that Axin2 and Snail are abundant in osteosarcoma patient samples, and both niclosamide and pyrvinium effectively induce the reversion of EMT by suppressing Axin2 and Snail in osteosarcoma cell lines. Considering the limited current treatment options for recurrent and metastatic osteosarcoma, our findings suggest potential therapeutics based on a novel EMT target for osteosarcoma patients.

## 2. Materials and Methods

### 2.1. Cell Culture and Reagents

Human osteosarcoma cell lines (MG-63 and SAOS-2) were obtained from the Korean Cell Line Bank (Seoul, Korea). MG-63 and SAOS-2 cells were cultured in RPMI 1640 (Lonza, Basel, Switzerland) with 10% FBS maintained at 37 °C in a humidified atmosphere containing 5% carbon dioxide. The human fetal osteoblastic hFOB 1.19 cell line was cultured according to established ATCC protocols. Mycoplasma infection was tested regularly using a PCR-based kit (MP0040; Sigma, St. Louis, MO, USA). siRNA duplexes directed against human Axin2 (sc-35087) were obtained from Santa Cruz Biotechnology (Santa Cruz, CA, USA). Niclosamide (2′,5-dichloro-4′-nitrosalicylanilide) was purchased from Cayman (Ann Arbor, MI, USA), and pyrvinium (6-(dimethylamino)-2-[2-(2,5-dimethyl-1-phenyl-1H-pyrrol-3-yl) ethenyl]-1-methyl-4,4′-methylenebis[3-hydroxy-2-naphthalenecarboxylate] (2:1)-quinolinium) was purchased from Sigma-Aldrich (Burlington, MA, USA). Niclosamide (Cayman, Ann Arbor, MI, USA) and pyrvinium pamoate (Sigma, St. Louis, MO, USA) were solubilized in DMSO for in vitro experiments.

### 2.2. MTT Assay and Colony Formation Assay

For the MTT assay, 2 × 10^3^ cells were seeded into 96-well plates and incubated at 37 °C and 5% CO_2_. Following this, the cells were treated with different doses of niclosamide (0.125–0.5 μM) or pyrvinium (0.01–0.1 μM). After 72 h incubation, the cells were washed once with PBS, supplemented with 100 μL of 1× MTT (3-(4,5-dimethylthiazol-2-yl)-2,5-diphenyl-tetrazolium bromide (Amresco, OH, USA, 0793-1G), diluted in RPMI, and incubated at 37 °C and 5% CO_2_ for 3 h. After removing the medium, 100 μL of DMSO was added to the shaker for 15 min, and the absorbance was measured at 570 nm. To measure the cell clonogenic survival capacity, 5 × 10^4^ cells were seeded in 6-well plates. After 3 days of exposure with various concentrations (0~0.5 μM) of the niclosamide or (0~0.1 μM) of pyrvinium, the cells were washed with PBS and cultured in normal culture medium for an additional 10–15 days to determine clonogenic survival. After crystal violet (0.5% *w*/*v*) staining, colonies of more than 50 cells were counted under a stereomicroscope. The number of colonies in 5 randomly chosen fields was determined under a high-power stereomicroscope. Cultures were photographed, and the number of colonies was counted.

### 2.3. Western Blot Analysis

We plated 2 × 10^5^ cells per well on 6-well plates. After 24 h of treatment with niclosamide and pyrvinium, cells were washed twice with PBS and placed on ice for 15 min with 1% Triton X-100 lysis buffer (50 mM Tris pH 8.0, 150 mM NaCl, 1 mM EDTA, 1% Triton X-100). After the reaction, the cells were collected using a scraper and centrifuged at 13,200× *g* for 15 min at 4 °C. The supernatant was collected and transferred to a new tube. Protein was quantified using the BCA Protein Assay (Thermo, Waltham, MA, USA), placed in 5× sample buffer, boiled for 10 min, and stored on ice. Electrophoresis was performed on an 8% SDS-polyacrylamide gel for Axin2 and E-cadherin and 12% SDS-polyacrylamide gel (Bio-Rad, Hercules, CA, USA) for Snail, which was then transferred to a nitrocellulose membrane (Whatman, Maidstone, UK). After transfer, the membrane was blocked in 5% skim milk (BD Biosciences, Franklin Lakes, NJ, USA) for 1 hour. The primary antibody was added and reacted at room temperature for 3 or 12 h at 4 °C. Primary antibodies against Snail (Cell Signaling, Danvers, MA, USA, L7062, 1:2000), β-catenin (BD, 610154, 1:2500), and α-Tubulin (AbFrontier, Seoul, Korea, LF-PA0146A, 1:5000) were used. Secondary antibodies were anti-rabbit IgG, HRP-linked (Cell Signaling, 7074S, 1:5000) and anti-mouse IgG, HRP-linked (Cell Signaling, 7076S, 1:5000). Detection was performed using WEST SAVE (AbFrontier, Seoul, Korea, LF-QC0101).

### 2.4. Reporter Gene Assay

A TOPFlash (plasmid 12456) reporter construct was obtained from Addgene (Watertown, MA, USA). Wt and E-box mutated E-cadherin reporter gene constructs (nt −108 to +125), Ecad(−108)-Luc and Ecad(−108)/E.BoxA.MUT/E.BoxB.MUT/E.BoxC.MUT-Luc were used as described previously [10,11]. For TCF/LEF and E-cadherin reporter assays, we plated 5 × 10^4^ cells per well on 12-well plates, and cells were transfected with 50~100 ng of Reporter DNA and 1 ng of SV40 Renilla (Promega, WI, USA, E6911) using Lipofectamine 2000 (11668-019, Invitrogen, Waltham, MA, USA). The cells were incubated with 0.25 μM niclosamide and 0.03 μM pyrvinium at 24 h after transfection. The cells were lysed after 24 h incubation. Luciferase and Renilla activities were measured using the Dual-Luciferase Reporter System Kit (Promega, Madison, WI, USA, E1910), and the luciferase activity was normalized against Renilla (Promega, Madison, WI, USA) activity. The results are expressed as averages of the ratios of reporter activities from triplicate experiments.

### 2.5. RNA Extraction and Real-Time PCR Analysis

For real-time quantitative PCR (qPCR), 2~5 × 10^5^ cells were used. Total RNA was isolated using TRIzol reagent (Invitrogen) following the manufacturer’s protocol. Reverse transcription premix (Intron, Seoul, Korea) was used to generate cDNA. qPCR for Axin2, Snail and β-catenin target genes and EMT-related genes was performed using an ABI-7300 instrument according to the SYBR Green Mix protocol (Takara, Shinga, Japan, RR82LR). The ΔCt value for each sample was calculated by normalizing against GAPDH. Primers used in this experiment are listed in Appendix A.

### 2.6. Transwell Assay

For migration assays with niclosamide and pyrvinium, osteosarcoma cells were seeded into transwell inserts (8.0 μm pore, BD Biosciences, Palo Alto, CA, USA). The filter inserts were prewetted before the cells were added. Cells were added to 5 × 10^4^/100 μL medium in the top of the inserts. The bottom chamber was filled with medium containing 0.25 μM niclosamide and 0.03 μM pyrvinium. After a culture period of 48 h for MG-63 cells and 72 h for SAOS-2 cells, the cells were washed twice with 1× PBS and fixed with 4% formaldehyde for 2 h. The upper part was wiped with cotton, and the cells in the lower part were stained with 0.25% crystal violet. Cell counts were determined in five random fields.

### 2.7. Tissue Samples and Immunohistochemistry (IHC)

Three paraffin-embedded tissue sections were obtained from surgically resected specimens diagnosed with osteosarcoma, including adjacent normal bone tissues of the same patients. All the specimens were provided by the Department of Oral Pathology, Yonsei University Dental Hospital. This study was approved by the Institutional Review Board for Bioethics of Yonsei University College of Dentistry (IRB 2-2018-0004). For histological and immunohistochemical examination, serial paraffin sections were stained with hematoxylin and eosin (H&E) for routine morphological observation. For the immunohistochemical study of Axin2 and Snail, tissue sections in microslides were deparaffinized in xylene and hydrated in gradually decreasing concentrations of ethanol. Following antigen retrieval, using citrate buffer (Dako, Glostrup, Denmark), the tissue sections were incubated with protein blocking agent at room temperature for 20 min, and then incubated with primary antibodies, monoclonal rabbit antihuman Axin2 (Abcam, Cambridge, UK, ab32197) (1:200) and polyclonal rabbit anti-human Snail antibody (1:2000) at room temperature for 2 h. The rabbit polyclonal antibody against Snail has been described previously [11]. After washing with PBS three times, the sections were incubated with second antibody, the Real Envision HRP Rabbit/Mouse Detection System (Dako, Carpinteria, CA). The chromogen was developed 3,30-diaminobenzidine followed by counterstaining with Meyer’s hematoxylin.

### 2.8. Statistical Analyses

Comparisons between two groups were performed with chi-square tests and independent *t*-tests. A *p*-value of less than 0.05 was statistically significant.

## 3. Results

### 3.1. Axin2 Could Be an Important EMT Target Candidate in Human Osteosarcoma

*Axin2* is a Wnt downstream target gene, and serves as a regulator of the Wnt signaling pathway [11]. The Axin2–GSK3β interaction site is an important therapeutic molecular target based on the EMT identified so far [11,24]. In order to determine the roles of Axin2 and Snail in clinical samples of osteosarcoma patients, we first detected the protein expression of both Axin2 and Snail by immunohistochemical study in the surgically resected osteosarcoma samples and adjacent normal bone tissues of same patents. There was no clear expression Axin2 and Snail in the osteocyte of normal bone tissues. Osteosarcoma sections presented highly co-expressed Axin2 and Snail compared to normal bone tissues. Within a series of human osteosarcoma tissues, Axin2 usually showed cytoplasmic expression, and *Snail* expression was mostly found in both the nucleus and cytoplasm of osteosarcoma cells in tumor lesions (Figure 1A). To determine the role of the Axin2 in osteosarcoma, we compared the mRNA levels of Axin2 in human osteosarcoma cell lines (MG-63 and SAOS-2) and human fetal osteoblastic hFOB 1.19 cells. Axin2 transcript levels were significantly increased in osteosarcoma cell lines, compared with the osteoblastic hFOB 1.19 cells (Figure 1B). Supporting the hypothesis that Axin2 and Snail show protein and transcript accordance, we evaluated Snail levels in the knockdown of Axin2. The knockdown of Axin2 showed the downregulation of endogenous protein abundance and the mRNA transcript level of Snail (Figure 1C,D). These findings strongly suggest that the hyperactivation of the Wnt pathway and the subsequent EMT program play important roles in the pathogenesis of osteosarcoma, and Axin2 and Snail may constitute the therapeutic targets of osteosarcoma.

### 3.2. The Non-Cytotoxic Levels of Niclosamide and Pyrvinium for Osteosarcoma Cells at nM Concentrations

While niclosamide and pyrvinium have emerged as therapeutics for various human cancers [18,19,20,21,22,23,24], their effect on osteosarcoma has not yet been determined. A previous study revealed that the non-cytotoxic nM concentration of niclosamide was sufficient to suppress the tumorigenic potential of colon cancer [24]. Therefore, we first analyzed the nM concentration at which niclosamide and pyrvinium inhibit the Wnt/EMT axis in osteosarcoma cell lines. To identify the non-cytotoxic concentration of niclosamide and pyrvinium, a colony formation assay and MTT assay were performed for osteosarcoma cells treated with niclosamide and pyrvinium. Osteosarcoma cells were treated with various nanomolar concentrations of niclosamide and pyrvinium for 72 h. After incubation for an additional 10~15 days, cells were stained with crystal violet (Figure 2A), and the number of colonies was counted. In both MG-63 and SAOS-2 osteosarcoma cells, there were no significant difference in the number of colonies at nM concentrations (Figure 2B). To determine the effect of niclosamide and pyrvinium on the inhibition of the Wnt/Snail axis at non-cytotoxic concentrations, we screened the cell death of osteosarcoma cells with 0.25 µM niclosamide and 0.03 µM pyrvinium. Both niclosamide and pyrvinium did not induce the cell death of osteosarcoma cells at the nM level (Figure 2C). From these results, we then conducted the following experiments at nM concentrations of niclosamide and pyrvinium.

### 3.3. Niclosamide and Pyrvinium Suppress the Canonical Wnt and Axin2-Dependent Snail-Mediated EMT at nM Levels

With reference to non-cytotoxic levels of niclosamide and pyrvinium in osteosarcoma cells of the results in Figure 2, we first examined the mRNA levels of Axin2 and Snail. Niclosamide and pyrvinium suppressed Axin2 and Snail transcript levels in two osteosarcoma cell lines, MG-63 and SAOS-2 cells, at nanomolar concentration, respectively (Figure 3A). Wnt signaling activates Axin2-dependent nuclear export of GSK3β and stabilizes β-catenin and Snail in the same manner via the inhibition of phosphorylation by GSK3β [10]. Then, we evaluated Axin2, β-catenin and Snail protein abundance after treatment with niclosamide and pyrvinium. Niclosamide and pyrvinium suppressed the Axin2 protein level. Both niclosamide and pyrvinium effectively suppressed the protein levels of β-catenin and Snail in osteosarcoma cell lines at nanomolar concentrations, respectively (Figure 3B). Wnt signaling inhibits the targeting of β-catenin and Snail for degradation, which results in the accumulation of β-catenin and Snail in the cytosol and nucleus. The accumulated nuclear β-catenin binds to the LEF/TCF transcription factor family and transcriptionally actives various target genes [31,32]. Then, we examined the TCF/LEF transcriptional activity with TOPFlash assay and abundances of ß-catenin/TCF-regulated target genes with quantitative PCR, after treatment of niclosamide and pyrvinium. Both niclosamide and pyrvinium effectively decreased TCF/LEF transcriptional activity (Figure 3C) and the expression of TCF/LEF target genes, including CD44, FGF18, ID2, EPHB2, EPHB3, MYCN, NOTUM, TCF4, and TCF7, in osteosarcoma cells at non-cytotoxic nM concentrations (Figure 3D). These findings suggest that nM levels of niclosamide and pyrvinium were sufficient to inhibit the Wnt/Axin2/Snail axis in osteosarcoma cell lines.

### 3.4. Niclosamide and Pyrvinium Effectively Induce the Reversion of EMT in Osteosarcoma Cells

The canonical Wnt pathway directly regulates cancer EMT programs via activation of the β-catenin/TCF complex-regulated Axin2/GSK3β/Snail cascade [10,11]. We next examined E-cadherin protein levels in response to treatment with niclosamide and pyrvinium, and found that the treatments of nM levels of niclosamide and pyrvinium increased E-cadherin protein abundances in osteosarcoma cells (Figure 4A). The transcription repressor Snail has been shown to bind directly to E-boxes in the E-cadherin promoter and to repress the transcription of E-cadherin [8,9]. Therefore, we transfected the E-cadherin promoter construct, Ecad(−108)-Luc, which contains the wild-type (WT) promoter sequence from nt −108 to +125 of the endogenous E-cadherin promoter or a control construct, Ecad(−108)/EboxA.MUT/EboxB.MUT/EboxC.MUT-Luc, which harbors mutations in all three E-boxes (3 × Mut) in both MG-63 and SAOS-2 cells. We found that both niclosamide and pyrvinium upregulated E-cadherin reporter activity at nanomolar concentrations (Figure 4B). During the EMT program of cancer cells, Snail downregulates not only E-cadherin, but also other epithelial markers [33,34,35]. On the contrary, mesenchymal markers are upregulated by Snail [36]. We examined the mRNA levels of EMT markers, both epithelial and mesenchymal. The treatments of niclosamide and pyrvinium suppressed the expression of mesenchymal markers. such as vimentin and fibronectin, and upregulated the expression of epithelial markers, such as E-cadherin and occludin (Figure 4C). These results indicate that both niclosamide and pyrvinium can reverse the EMT process in MG-63 and SAOS-2 osteosarcoma cell lines at non-cytotoxic concentrations.

### 3.5. Niclosamide and Pyrvinium Effectively Inhibit the Migration of Osteosarcoma Cells

The Wnt signaling pathway directly engages Snail-mediated EMT programs, and it is involved in many events during cancer progression, such as therapeutic resistance and metabolic reprogramming as well as invasion and metastasis [3,37,38]. We investigated the anti-EMT effects of niclosamide and pyrvinium with transwell migration assays. The numbers of migrated cells were lower in the cells treated with niclosamide and pyrvinium than those in the DMSO-treated control group. We also demonstrated that the treatments with 0.25 µM niclosamide and 0.03 μM pyrvinium for 24 h significantly inhibited the migration of osteosarcoma cells, respectively. There was an 80% (*p* < 0.01) decrease in the number of migrated SAOS-2 cells treated with niclosamide and pyrvinium compared to the control cells (Figure 5).

## 4. Discussion

Epithelial to mesenchymal transition (EMT) is essential in morphogenesis during embryonic development, tissue repair, organ fibrosis and malignant transformation of tumors, such as invasion and metastasis. EMT also confers cells with stem cell-like properties, which is triggered by the coordinate control of gene regulatory networks, including transcriptional and translational regulation [7,39,40], as well as the regulation of protein stability by post-translational modification [10,41,42]. Due to the transient nature of the EMT program, it was generally considered unsuitable as a therapeutic target. With the results that inhibitors of cancer stem cells are approximately ten times more effective than previous anticancer drugs in the in vivo suppression of metastasis [43], EMT is finally in the spotlight as a target for next-generation cancer diagnosis and therapeutics. An emergent therapeutic target based on EMT is the Axin2–GSK3β interaction site [11]. In this study, we found that Axin2 and Snail are co-localized in invasive cancer cells within human osteosarcoma tissues and Axin2 is activated in osteosarcoma cell lines compared to human fetal osteoblasts. These findings strongly suggest that the hyperactivation of Wnt signaling and the subsequent EMT program contributes to malignant conversion, such as the invasion and metastasis of osteosarcoma, and Axin2 and Snail are candidate novel EMT targets for human osteosarcoma. As the prognosis is very poor for patients with metastatic or recurrent osteosarcoma, the novel EMT target related to tumor recurrence and metastasis will be of great help in the discovery of novel therapeutic targets and the development of predictive biomarkers for osteosarcoma patients.

The significance of EMT during cancer progression has been emphasized, but its precise regulatory mechanism remains undiscovered. The regulatory machinery controlling reprogramming of cells during cancer progression has been recently elucidated. Emerging evidence suggests that the EMT program may be necessary for cancer cells not only to invade, but also to resist apoptosis and survive during cancer progression [44,45,46,47]. The EMT inducer Snail promotes cancer cell survival under metabolic stress during cancer progression by reprogramming the energy metabolism [37,38]. The protein stability of Snail is regulated by GSK3β-dependent phosphorylation, β-TrCP-directed ubiquitination and proteasomal degradation in the same manner as β-catenin [42]. The Wnt signaling pathway stabilizes both β-catenin and Snail simultaneously via the inhibition of phosphorylation by GSK3β [11]. Previous studies have proposed Axin2 as a negative regulator of the Wnt pathway as Axin2 acts as a chaperone for GSK3β and promotes GSK3β-dependent phosphorylation in the cytoplasm [13,14]. However, Axin2 has been demonstrated to play an oncogenic role through its nuclear export function of GSK3β, resulting in the nuclear stabilization of Snail and induction of the EMT program [11]. The oncogenic role of Axin2 is also supported by the fact that Axin2 mediates GSK3β-dependent phosphorylation and the activation of LRP6, a Wnt co-receptor at the plasma membrane [48,49]. In this study, we revealed that niclosamide and pyrvinium, known as Wnt inhibitors, effectively target Axin2 and suppress Snail-mediated EMT in osteosarcoma cells. These results suggest that both niclosamide and pyrvinium could be used as novel EMT therapeutics for patients, especially with metastatic or recurrent osteosarcoma.

Despite the improved survival rate of chemotherapy with current cytotoxic chemotherapeutic drugs for osteosarcoma, pediatric osteosarcoma has the lowest survival rates, with the best reported 10-year survival rate of 92% [50]. Therefore, the development of new chemotherapeutic drugs with novel modes of action is essential. In this study, we found that niclosamide and pyrvinium, FDA-approved safe anthelmintic drugs, do not induce the cell death of osteosarcoma cells at nanomolar concentrations, but induce the reversion of EMT at the same concentrations. Our study provides a novel repositioned therapeutic option for osteosarcoma patients by targeting EMT. Further studies of dosage with in vivo efficacy are needed to confirm the therapeutic potential and clinical benefit in patients with osteosarcoma.

## 5. Conclusions

Axin2 and Snail are candidate therapeutic targets based on EMT for human osteosarcoma. Additionally, the anthelminthic agents niclosamide and pyrvinium, targeting Axin2, may be effective novel EMT therapeutic drugs for patients with osteosarcoma.

## Figures and Tables

**Figure 1 cancers-13-04630-f001:**
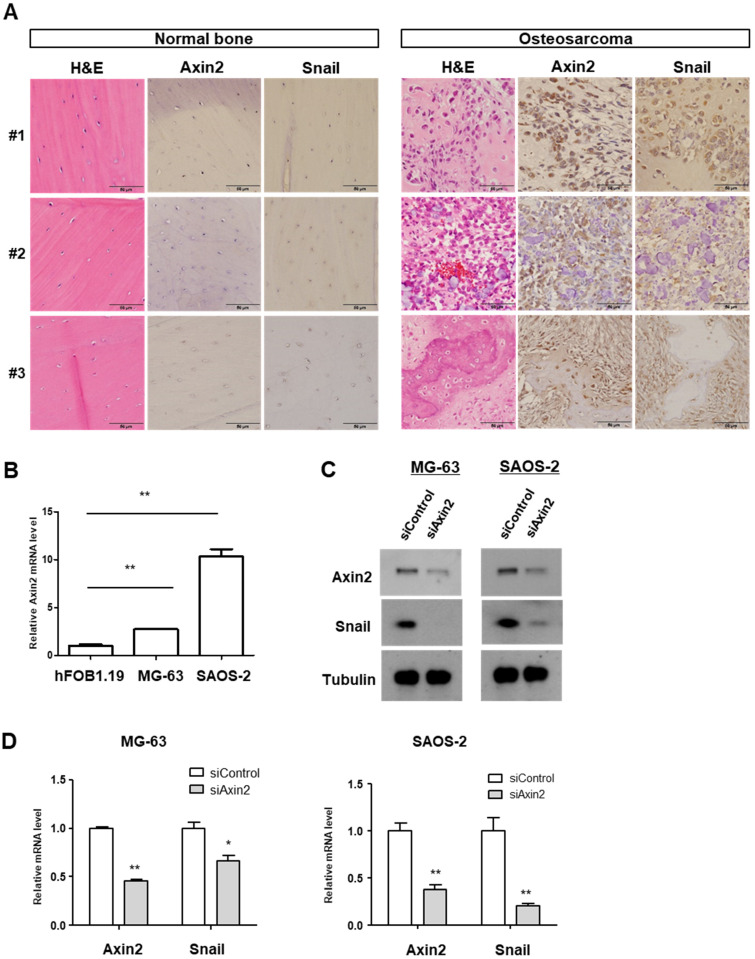
Axin2 and Snail are highly expressed in clinical tissue samples and cells of osteosarcoma. (**A**) Routine H&E staining and Axin2 and Snail immunohistochemical staining in adjacent normal bones (let panel) and tumor samples (right panel) in three patients with osteosarcoma. Photomicrographs were obtained at ×400 magnification. Scale bar = 50 µm. (**B**) The Axin2 mRNA levels of osteosarcoma cell line (MG-63, SAOS-2). The level was presented as the fold change relative to that of the hFOB1.19 cells. Quantitative RT-PCR was carried out in triplicate, at the minimum (error bars, ± SD. ** *p* < 0.01 versus hFOB1.19). (**C**) Axin2 and Snail protein levels of osteosarcoma cell treated with niclosamide and pyrvinium by using Western blotting. Representative images are shown from at least two independent experiments. For uncropped figures, see Appendix A. (**D**) The Axin2 and Snail mRNA level in Axin2 knockdown cells. Cells were transfected with scrambled control and siAxin2. Axin2 and Snail transcript abundance was analyzed with quantitative RT-PCR. Quantitative RT-PCR experiments were carried out in triplicate, at the minimum (error bars, ± SD. * *p* < 0.05, ** *p* < 0.01 versus siControl).

**Figure 2 cancers-13-04630-f002:**
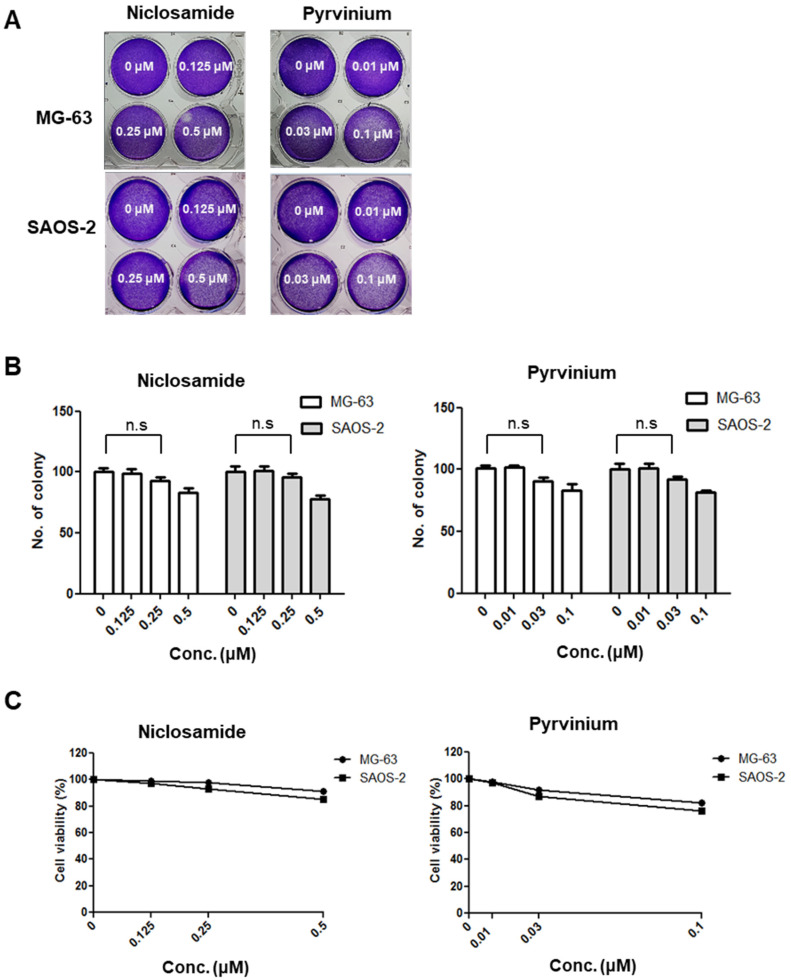
Niclosamide and pyrvinium do not induce cell death at non-cytotoxic nM concentration in osteosarcoma cells. (**A**) Representative images of colonies in MG-63 and SAOS-2 cells against niclosamide and pyrvinium at the indicated concentration. Osteosarcoma cells treated with 0.125, 0.25, and 0.5 μM niclosamide and 0.01, 0.03, and 0.1 μM pyrvinium for 72 h and then stained with 0.25% crystal violet after colony formation. (**B**) The colony number of osteosarcoma cells treated with niclosamide and pyrvinium. Colonies of more than 50 cells were counted after crystal violet staining. For all of the above assays, each assay condition was evaluated in triplicate. Results are expressed as the mean ± SD (error bars; *n* = 3, n.s., not significant). (**C**) The survival curve of osterosarcoma cells treated with niclosamide and pyrvinium for 72 h.

**Figure 3 cancers-13-04630-f003:**
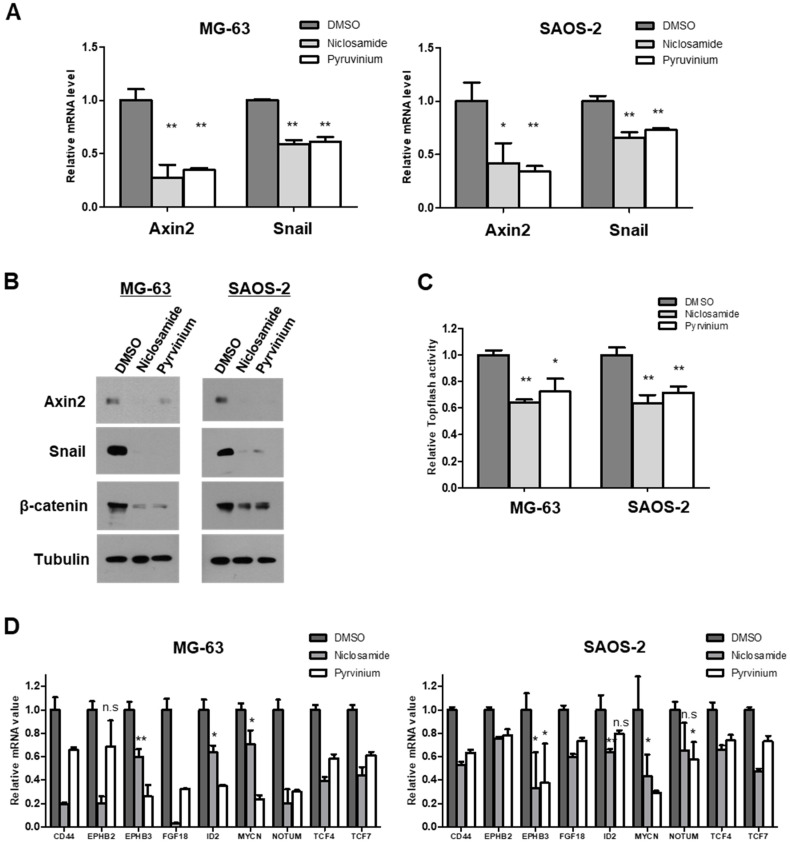
Niclosamide and pyrvinium attenuate the activities of Wnt/Axin2/Snail axis. (**A**) Axin2 and Snail mRNA level in MG-63 and SAOS-2 cells. The transcript abundance was determined by quantitative RT-PCR. (**B**) Western blot analysis of osteosarcoma cells treated with 0.25 µM niclosamide and 0.03 µM pyrvinium. Tubulin served as a loading control. Representative images are shown from at least two independent experiments. (**C**) Relative luciferase activity representing β-catenin/TCF-dependent transcription in treated cells. Osteosarcoma cells were transiently transfected with the TOPFlash luciferase construct and SV40 Renilla-expression vector in each well. After 24 h of incubation, cells were treated with 0.25 µM niclosamide and 0.03 µM pyrvinium. Luciferase activity was measured 24 h later with normalization to the activity of SV40 Renilla. (**D**) mRNA levels of β-catenin target genes by quantitative RT-PCR. The value of sample was normalized to the levels of GAPDH mRNA. Quantitative RT-PCR experiments were carried out in triplicate (error bars, ±SD. * *p* < 0.05, ** *p* < 0.01 versus DMSO, n.s., not significant).

**Figure 4 cancers-13-04630-f004:**
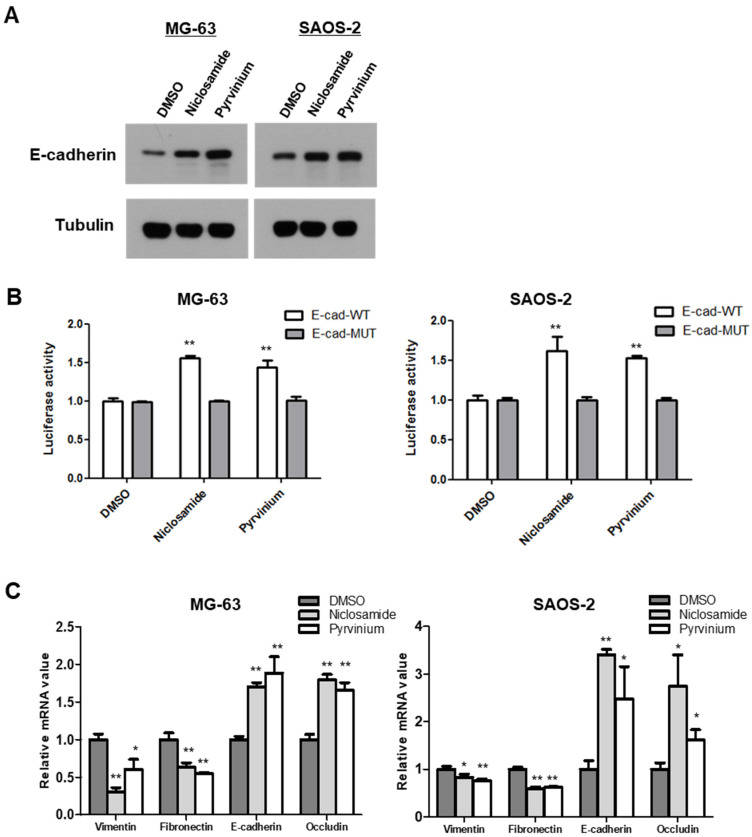
Niclosamide and pyrvinium increase the abundance of E-cadherin and potentiate cell-to-cell adhesion. (**A**) E-cadherin protein levels of osteosarcoma cell treated with niclosamide and pyrvinium by using Western blotting. Representative images are shown from at least two independent experiments. (**B**) The relative E-cadherin promoter activities. E-cadherin promoter activity of wild-type and E-box mutant constructs were determined in osteosarcoma cells. Induced luciferase activity was measured. (**C**) Quantitative RT-PCR analysis of EMT marker. The expression of EMT-related genes in MG-63 and SAOS-2 cells treated with niclosamide and pyrvinium was determined by quantitative RT-PCR. Relative mRNA expression levels were normalized to GAPDH. Quantitative RT-PCR experiments were carried out in triplicate (error bars, ±SD. * *p* < 0.05, ** *p* < 0.01 versus DMSO).

**Figure 5 cancers-13-04630-f005:**
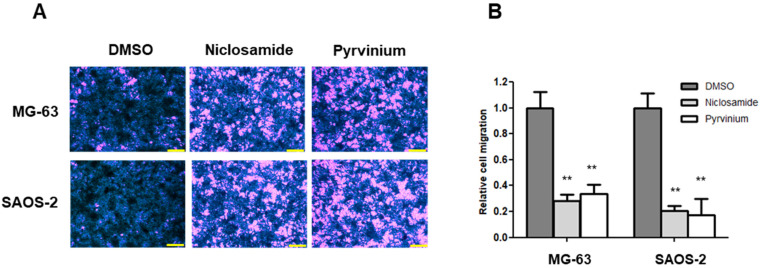
Niclosamide and pyrvinium inhibit the migration of osteosarcoma cells at non-cytotoxic μM concentration. (**A**) Representative migration images of MG-63 and SAOS-2; scale bar, 200 μm. (**B**) The migratory activity of MG-63 and SAOS-2 cells treated with 0.25 µM niclosamide and 0.03 μM pyrvinium using transwell migration assay. Results are representative of five independent experiments (error bars, ±SD. ** *p* < 0.01 versus DMSO).

## Data Availability

The data presented in this study are available in this article and the Appendix A.

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
