# Peer review of "Niclosamide and Pyrvinium Are Both Potential Therapeutics for Osteosarcoma, Inhibiting Wnt–Axin2–Snail Cascade"

_cancers, 2021, doi:10.3390/cancers13184630_

Round 1
Reviewer 1 Report
I would like to thank the authors for incorporating my suggestions. Best of luck with the next steps of your research!
This manuscript is a resubmission of an earlier submission. The following is a list of the peer review reports and author responses from that submission.
Round 1
Reviewer 1 Report
Please see attached file

Author Response
We separately attach our point-by-point detailed response to the reviewer’s comments.

Reviewer 2 Report
WIth this study, the authors proposed the pathway involving Axin2 and Snail as potential targetable target for treating osteosarcomas. Moreover, they showed the impact of niclosamide and pyrvinium in osteosarcoma at low toxicity concentrations. The two compounds reduced Axin2 and Snail levels and WNT pathway activation. This phenotype resulted in the inhibition of migration of osteosarcoma cells, and the authors proposed the two compounds as potential therapy for osteosarcoma. The study is well conducted and potentially interesting for the readers, however some additonal experiments are needed to increase the relevance of the obtained results.
Specifically, there are some issues that need to be addressed.
- In Figure 1A the authors analyzed the expression of Axin2 and Snail proteins by IHC in osteosarcoma patients. The authors stated, I quote their words "The protein of Axin2 and Snail 178 denotes higher relative expression significantly in immunohistochemistry". But "higher" relative to what? In respect to normal cells? To adjacent normal tissue? To non-invasive osteosarcomas? Given Figure 1A shows only the expression of these two proteins in the cancer tissue, this conclusion needs to be explained better. Moreover, 3 patients is a relatively limited number to draw some conlusions.
- In Figure 1C the authors showed that Axin2 and Snail mRNA levels are somehow linked; when Axin2 is knocked-down, Snail mRNA is down-regulated. Is this reduction influencing WNT pathway? It would be also importnat to show also these differences at the protein levels via WB upon siAxin2.
- In Figure 2A the authors should measure the mRNA levels of Snail given they demonstratwed that also the protein levels of Snail are down-regulated by the two compounds. Moreover, it would be important to show the decrease also of Axin2 protein by WB by the two compounds.
- Why were the migration tests conducted with 0.1uM pyrvinium instead of the selected 0.03uM? At this concentration it became toxic particularly in SaOs2 cells. However, this is only mentioned in the text. In the legend it was stated 0.03uM. Please cross-check this.
Minor issues:
- There are several typos throughout the manuscript (i.e., CO2 not subscript, "sere" instead of "were", etc). Please adjust them all.
- In the first paragraph of the methods: what is hFOB 1.19 cell line? It was not cited what cell type they are; it Is mentioned later, but it needs to be stated also here.
- Paragraph 2.4: Some details are missing. In what format were the gene reporter assays performed? 96-24-12-well plates? How many cells were seeded? The questions came given the really low amount of reporter (50ng?) and renilla (1ng?) plasmids used.
- The title of paragraph 3.1 is not ok. This conclusion is too bold. How could Axin2 be "THE" EMT candidate target? It might be one of the putative candidates for osteosarcoma, because many other factors play a significant role in osteosarcoma. The authors have to soften this title.
- In the legend of Figure 1, panel "C" has to be written as a bold character.
- Paragraph 3.2 (end of page 5): The time point indicated in the text for colony formation assays is wrong. How could it be done 24 hours after treatments? In the methods and in the legend it seems clear the treatment was for 72hours and cells were stained after a couple of weeks. Please revise this.
- Figure 2: All the captions for micromolar concentrations have to be corrected. The authors should use the Greek character for micro.
Author Response

(The authors gave the same response as above.)
